# Gradient-adjusted Incremental Target Propagation Provides Effective Credit Assignment in Deep Neural Networks

Sander Dalm   Nasir Ahmad   Luca Ambrogioni   Marcel van Gerven
**Department of Artificial Intelligence**
**Donders Institute for Brain, Cognition and Behaviour**
**Radboud University, Nijmegen, the Netherlands**

Reviewed on OpenReview: `https://openreview.net/forum?id=Lx19EyKX77`

## Abstract

Many of the recent advances in the field of artificial intelligence have been fueled by the highly successful backpropagation of error (BP) algorithm, which efficiently solves the credit assignment problem in artificial neural networks. However, it is unlikely that BP is implemented in its usual form within biological neural networks, because of its reliance on non-local information in propagating error gradients. Since biological neural networks are capable of highly efficient learning and responses from BP trained models can be related to neural responses, it seems reasonable that a biologically viable approximation of BP underlies synaptic plasticity in the brain. Gradient-adjusted incremental target propagation (GAIT-prop or GP for short) has recently been derived directly from BP and has been shown to successfully train networks in a more biologically plausible manner. However, so far, GP has only been shown to work on relatively low-dimensional problems, such as handwritten-digit recognition. This work addresses some of the scaling issues in GP and shows it to perform effective multi-layer credit assignment in deeper networks and on the much more challenging ImageNet dataset.

## 1 Introduction

Backpropagation of error (BP) is a learning algorithm that solves the credit assignment problem in deep neural networks, allowing for the formation of task-relevant internal representations and high performance in applications (Rumelhart, Hinton, and RJ, 1986; LeCun, Bengio, and Hinton, 2015; Schmidhuber, 2014). Despite its efficacy, the default construction of BP does not appear a likely candidate for the computational steps involved in the learning algorithm of real neural systems. Early criticisms of BP's biological plausibility have already been put forward by Grossberg (1987) and Crick (1989) in the late 1980's. A recent review by Lillicrap et al. (2020) provides a modern summary of the mechanisms that make BP biologically implausible. These include issues of symmetric synaptic weight matrices, error propagation machinery and more. In particular, these issues all pertain to the use of non-local information for the propagation and computation of error signals to update individual synaptic connections deep within a network. Therefore, biologically plausible learning algorithms are required to provide sensible methods for the assignment of credit deep within a neural network model using local-only information.

Several lines of previous work have attempted to address the biological implausibilities inherent to BP. Lillicrap et al. (2016) report the counter-intuitive result that a fixed random matrix of feedback weights can support learning in neural networks. This learning is presumed to take place as a result of the forward weight matrices adapting themselves to act as local pseudoinverses of the feedback matrices around the error manifold. This approach is called feedback alignment (FA) and it does away with the necessity of symmetric weight matrices. Direct feedback alignment (DFA) was proposed by Nøkland (2016) and works by connecting random feedback weights from the output layer directly to each hidden layer. This approach has been used to

train modern deep learning architectures (Launay et al., 2020), but its performance compared to regular FA on challenging datasets actually seems to be slightly worse (Bartunov et al., 2018). This may have to do with the relatively simplistic way of performing credit assignment, which does not fully exploit the expressiveness of multi-layer architectures.

Equilibrium Propagation (EP) (Scellier & Bengio, 2017) is a two phase energy-based model. In the first phase, the network's input units are clamped to a particular input and the rest of the network settles to a certain equilibrium state. In the second phase, the network's output units are weakly clamped, meaning that they are nudged toward a desired network output using a small constant clamping factor. This perturbation then propagates backward throughout the network and is used to inform weight updates, following a contrastive Hebbian learning rule. These updates are shown to follow the error gradient as they would have been computed by BP. An extension of this work applied EP, with some improvements, to convolutional neural networks to show performance comparable to BP on CIFAR10 (Laborieux et al., 2021). Though theoretically appealing, one downside appears to be that EP's energy dynamics come with significant computational overhead, with a CIFAR10 run reportedly taking two days on a GPU.

Burstprop (Payeur et al., 2021) is a learning rule for artificial pyramidal neurons that induces Long-Term Potentiation (LTP) if a presynaptic event is followed by a postsynaptic burst and Long-Term Depression (LTD) otherwise, where the bursting is controlled by topdown feedback connections. This method is shown to effectively learn on the ImageNet dataset and to approximate gradient descent on the error landscape (Greedy et al., 2022). A drawback of the method is that it simulates ensembles of neurons rather than individual neurons and demands a rather complex network topology with learned feedback connections and multi-compartment neurons.

Predictive coding (Whittington & Bogacz, 2017), alternatively, handles error signals by assuming the existence of separate error neurons for each value neuron. These neurons compute differences between predicted and observed activations and these prediction errors can be used to inform local weight updates. The presence of an error neuron for each value neuron, however, is a rather strong assumption without any neurobiological evidence supporting it. Though most predictive coding research has focused on smaller datasets (Millidge et al., 2022), Han et al. (2018) report success in learning CIFAR10 and ImageNet using a bidirectional predictive coding network.

Ideally, a biologically plausible algorithm would learn directly from changes in activities of neurons. An algorithm which aims to do just that is target propagation (TP) (Bengio, 2014; Lee, Zhang, Fischer, and Bengio, 2015). The key principle is to combine a desired network output with an (approximate) inverse model of a network in order to produce desired outputs for each layer of a network. These computed layer-wise target activities can then be taken as local supervised labels and used for learning. TP addresses the weight transport problem, because it uses only local activity targets to achieve multi-layer learning. However, there is no explicit theoretical relationship between TP and BP. Furthermore, it has been shown that the efficacy of TP in deep networks that are trained to solve difficult tasks is questionable (Bartunov et al., 2018).

Bartunov et al. (2018) show that biologically plausible algorithms like TP, FA and DFA perform reasonably well, though still worse than BP, on MNIST (Deng, 2012) and CIFAR10 (Krizhevsky, 2012) when applied to fully connected architectures. However, in locally connected architectures and on the ImageNet (Russakovsky et al., 2014) dataset, a significant performance gap appears between all biologically inspired algorithms and BP. Particularly, TP and its variations fail to learn almost completely on ImageNet. FA learns the task to an extent, but its final top-1 test accuracy is about one fourth that of BP.

A common shortcoming among the above approaches is that no theoretical guarantees are made that their updates closely approximate the error gradient as computed by BP. Biologically plausible approaches tend to use local learning rules to avoid the weight transport problem. These local targets often provide useful feedback information in relatively shallow networks, but when multi-layer credit assignment is required, compounding errors drive the updates away from those computed by BP. This failure to accurately track the error gradient likely explains why no activity based (e.g. TP) learning algorithm has so far managed to scale to ImageNet and remain competitive with BP in terms of performance. Scaling biologically plausible learning to real world problems calls for a learning rule that works based on local information but somehow still guarantees a close correspondence to BP.

Though TP in its original formulation does not track BP's weight updates very accurately, Ahmad et al. (2020) recently found that there exists a direct correspondence between target-based learning and BP, though this relationship only exists locally and under specific network constraints. The correspondence between target-based learning and BP can be arrived at through a method called gradient-adjusted incremental target propagation (GP). GP was derived specifically to maintain exact correspondence to BP in the presence of orthogonal weight matrices and exact inverses, which may themselves not be biologically plausible. However, the main contribution of GP is that it allows for a local, activity-based learning rule that still closely approximates BP. This is accomplished by computing those layer-wise targets which would produce weight updates equivalent to BP. In practice, GP was shown to match the performance of BP on the MNIST, FMNIST and KMNIST datasets for relatively shallow networks. However, deeper networks and more challenging tasks were not explored.

The current paper shows that GP in its standard form can diverge from BP due to issues of precision and proposes a solution to these issues. In particular, we propose a layer-wise target normalisation procedure that stabilizes learning with GP in deeper networks. We also describe an easy to implement procedure for inverting convolutional neural network layers in order to efficiently extend GP's utility to problems like the ImageNet classification task (Russakovsky et al., 2014).

## 2 Methods

### 2.1 Target propagation and GAIT propagation

GP is based on the idea of target propagation (Bengio, 2014; Lee et al., 2015). The key idea is that layer-wise activity targets, rather than error gradients, are propagated backwards through the network. The difference between the current activity and target activity acts as a local error signal to compute weight updates. An inverse mapping from layer $l$ to layer $l-1$ can be used to determine which activation vector in layer $l-1$ would have produced this more desirable activation in layer $l$, which then becomes the target for layer $l-1$, and so on. As discussed, though elegant, this method does not produce weight updates that are similar to those produced by BP. The first reason why TP yields different weight updates from BP, is simply because TP uses either learned or exact matrix inverses to propagate its activity targets, while BP uses the transpose of the forward matrix to propagate its errors backwards. For non-orthogonal weight matrices, the inverse differs from the transpose. Furthermore, TP does not account for local gradients when perturbing the current activities in the direction of target activities.

GP seeks to address both of these problems. In order to ensure that matrix inverses produce the same transformation as matrix transposes, it constrains weight matrices to be orthogonal. This is achieved by orthogonal initialization of weights, followed by regularization during each weight update. GP also accounts for local gradients by adjusting its activity perturbations, multiplying them by the square of the local gradient. Under the constraint of square orthogonal weight matrices, GP follows the gradient as computed by BP arbitrarily closely and performs competitively with BP on datasets on the scale of MNIST. In this paper, we find that when scaling up GP to more complex datasets, and thus larger networks, numerical instability issues occur. The resolution of this scaling issue is the first major subject of this paper.

Let us define the function $F$ as the forward-pass mapping between layers in a neural network such that for layer $l+1$,

$$a_{l+1} = F_l(a_l) = f(W_l a_l + b_l) \tag{1}$$

where $a_l$ is the activation vector, $W_l$ is the weight matrix and $b_l$ is a bias vector for layer $l$. The function $f(\cdot)$ represents an activation function. In TP, layer-wise targets, $t^{\mathrm{tp}}$ are propagated backwards from layer $l$ to layer $l-1$, by taking the targets of layer $l$ and applying a (learned) inverse function to them:

$$t_{l-1}^{\mathrm{tp}} = G_l\left(t_l^{\mathrm{tp}}\right) \tag{2}$$

where $G_l$ is the (learned) inverse mapping from layer $l$ to layer $l-1$ so that with a perfect inverse mapping, $G_l(F_l(a_l)) = a_l$. Exact inverses require an equal number of neurons in each layer, however, this constraint can be relaxed by using auxiliary units which can be interpreted as a form of perceptual memory Ahmad

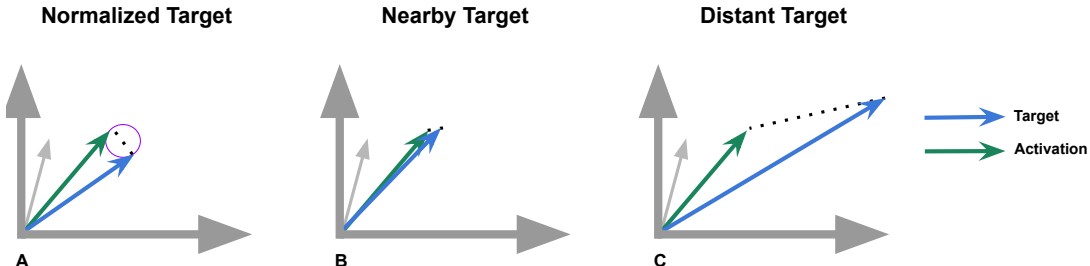

Figure 1: A depiction of the activation and target vectors. A) Targets and activations are normalized to be at a specific distance. B) Targets and activations become too close such that floating point precision issues occur. C) Targets and activations diverge, weakening the correspondence between GP and BP.

et al. (2020). In this work (and in the original GP work), the inverse function $G_l$ is defined exactly such that

$$a_l = F_l^{-1}(a_{l+1}) = G_l(a_{l+1}) = W_l^{-1}(f^{-1}(a_{l+1}) - b_l) \tag{3}$$

In order to enable this exact inversion, weight matrices are square and initialized as orthogonal and the Leaky-ReLU activation function (invertible for all real-valued outputs) was used.

GP proposes a modification to the targets being inverted. The inverted targets were modified such that they constitute a small perturbation from the forward-pass activity toward the target activity. The perturbation is also multiplied by the square of the activation function derivative at the forward-pass activity. Given this definition, the GP targets, $t^{\mathrm{gp}}$, are computed after a transformation such that:

$$t_{l-1}^{\mathrm{gp}} = G_{l-1}\left((I - \epsilon_l)\ a_l + \epsilon_l\ t_l^{\mathrm{gp}}\right) \tag{4}$$

where $\epsilon_l = \gamma_l D_l^2$ is a perturbation parameter with $\gamma_l$ a constant with very small magnitude and $D_l$ a diagonal square matrix with the derivatives $\frac{dF_l}{da_l}$ of the forward mapping $F_l$ on its main diagonal. This target adjustment is constructed in order to obtain updates equivalent to BP under the condition of orthogonal weight matrices Ahmad et al. (2020).

## 2.2 Scaling GP to ImageNet

Scaling GP to problems like ImageNet requires us to address the numerical instability that arises in GP when it is applied to deeper networks, as described in the next section. In addition, it requires the implementation of local connectivity to make parameter estimation feasible. Recall that it was the application of deep convolutional neural networks (CNNs) to the ImageNet challenge in 2012 that sparked the revival of neural networks for image classification (Krizhevsky et al., 2012). To our knowledge, there are no practical methods for achieving high performance on ImageNet without using local connectivity. To address these challenges, we propose a method of normalizing target-activation distances and a method to utilize an invertible convolution operation using GP, which sidesteps the need to invert large matrices in memory.

### 2.2.1 Normalized targets

We find empirically that when inverting targets deep in a network, precision errors in the inversion can occur when the target versus activity difference becomes very small. Repeated inversion with fixed layer-wise incremental factors $\gamma_l$ can result in a net increase or decrease (explosion or vanishing) of the target-activation distance, which is the key error term used for learning.

To alleviate the problem of target-activation differences becoming susceptible to precision errors (or increasing beyond the limits of our linear approximation), we propose a local normalization of the incremental factors $\gamma_l$ based upon the target-activation difference (Figure 1). For each layer independently, we divide the incremental factor by the L2-norm of the target-activation difference $\gamma_l$. This ensures that our 'distance to target' is fixed

on a layer-wise basis, ensuring robust error propagation. Specifically, the value of the incremental factor for layer $l$ is computed as

$$\gamma_l = \frac{\eta}{\|a_l - t_l^{\text{gp}}\|^2} \tag{5}$$

where $\eta$ is a normalization constant determining the desired Euclidean distance between targets and activations. Note that when target-activation distances are very small, $\gamma$ may be bigger than 1. This amounts to inflating the target-activation distance rather than decreasing it. See Appendix B for a pseudocode explanation of GP's backward pass which includes normalized targets.

### 2.2.2 Application of GP to CNN architectures

GP requires network layers to be invertible. Because inverting large sparse matrices in memory is prohibitively expensive, we introduce a way to invert convolutional layers in a patch by patch manner, greatly reducing computational cost. The output of a convolutional layer is inverted by running a transposed convolution that takes as its input the layer's output and as its weight matrix the same kernel as used in the forward pass, but inverted. For these four-dimensional kernels to be inverted, they first need to be reshaped to be square, then transposed, then inverted, then reshaped back to their original format. See Appendix B for a pseudocode explanation. Note that if these kernels were always perfectly orthogonal, simply taking the transpose would be equivalent to inversion. However, since our regularizer does not ensure perfect orthogonality, we take the exact inverse.

The above method correctly inverts a convolutional layer's output only in the case where receptive fields do not overlap. When receptive fields do overlap, those parts of the input image that are in the scope of multiple receptive fields will be reconstructed multiple times, requiring a renormalization. An easy way to accomplish this, is to divide each pixel in the reconstruction by the number of receptive fields it is a part of. This will correctly reconstruct any input image.

However, this process will also divide error information flowing through the pixel by the same number, causing a reduction in the error signal. To remedy this in the context of GP, we use a different method: the input reconstruction is corrected by removing from it the forward signal (rather than feedback signal) equivalent to the number of reconstructions (minus one). This allows the layer to reconstruct its input exactly once. Importantly, any error information coming from higher layers will be preserved in this way such that error information is fed back and integrated from multiple overlapping receptive fields. See Appendix C for a pseudocode explanation.

### 2.2.3 Additional constraints

Exact inverses can only be computed for full-rank square matrices in which there is no loss of dimensionality. This means that, at minimum, the number of output channels of a filter must be equal to the input dimensionality, that is, the number of output channels must be equivalent to the kernel height times kernel width times number of input channels ($C_{\text{out}} = C_{\text{in}} \times H \times W$).

In order to achieve a change in network width, auxiliary output channels can be used akin to the auxiliary units used by Ahmad et al. (2020). This amounts to slicing off a number of channels from the output of a convolutional layer, before feeding the result into the next layer. When inverting the convolutional layer, its entire output, including the auxiliary units, should be used as input to the inversion operation. The auxiliary outputs should therefore not be discarded after the forward pass, but kept in memory.

### 2.3 Network Models and Training

To investigate the above two proposed improvements, together referred to as normalized GP, we trained and tested several deep neural network models with the CIFAR10 and ImageNet datasets. We trained two fully connected networks on CIFAR10 using either BP, FA, GP or normalized GP; one shallow (three hidden layers) and one deep (six hidden layers). We further trained a convolutional network on ImageNet using only normalized GP, FA or BP. FA was chosen as an additional bio-plausible benchmark because it performed the best of all bio-plausible algorithms in Bartunov et al. (2018), though it still lagged significantly behind BP.

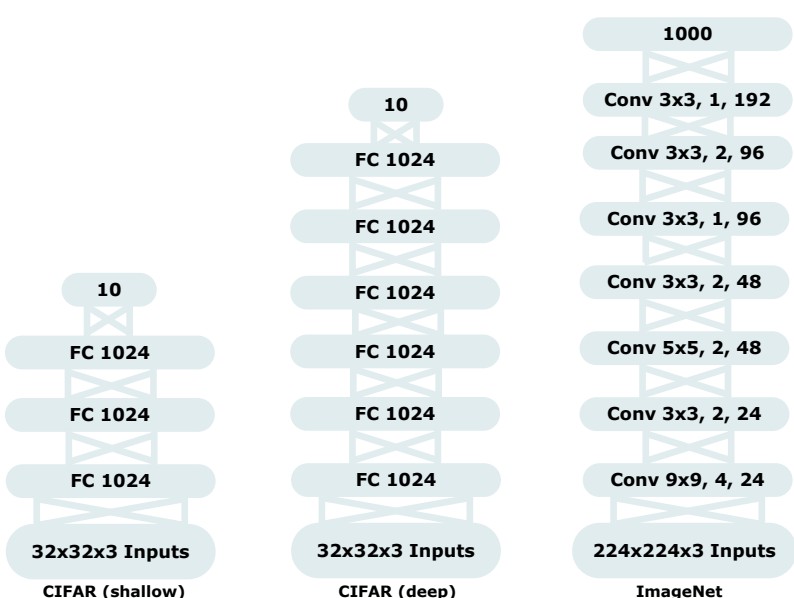

Figure 2: Architectures used to train on CIFAR10 shallow (left), CIFAR10 deep (middle) and ImageNet (right). Fully connected layers are indicated with FC, convolutional layers with Conv. For FC layers, the number indicates the number of units. For Conv layers, the numbers represent the kernel size, stride and output channels, respectively.

The convolutional architecture and hyper-parameters were also adapted from Bartunov et al. (2018). Due to memory constraints, we reduced the number of filters by half when training ImageNet. This likely impacts performance, however we desired to investigate whether GP can perform competitively with BP on a large dataset in a deep convolutional network, rather than to maximize peak performance.

During training, parameters were updated for all methods using the Adam optimizer (Kingma & Ba, 2014). In addition, when using GP, a regularizer was used to encourage orthogonality of weight matrices. The strength of the regularization component is a scalar hyper-parameter that indicates by what number the regularization component is multiplied before adding it to the weight update. See Ahmad et al. (2020) for details. In addition to being regularized to be orthogonal, all weight matrices were also initialized to be orthogonal, to ensure efficient training from the start of the experiment. Other performance enhancing techniques like batch normalization, L2-regularization and dropout were not used. All network layers implemented the leaky-ReLU activation function and categorical cross-entropy was used as a loss function.

## 3 Results

In the following, we consider networks trained on CIFAR10 and ImageNet using different learning algorithms. Appendix A provides an overview of the train and test accuracies at the end of training. We will use vGP for the vanilla GP formulation as used in Ahmad et al. (2020) and reserve GP for the formulation which uses normalized targets as proposed in this paper.

### 3.1 CIFAR10 results for a shallow network

The correspondence between weight updates from different algorithms was measured by treating each update as a one dimensional vector and measuring the angle between them. Figure 3A and B show the angles between (v)GP's and FA's weight updates with respect to those of BP when training a relatively shallow fully connected network on CIFAR10. The angle between the updates fluctuates around one degree for (v)GP, indicating a very high degree of correspondence between the algorithms. FA's updates show angles between 60 and 80 for all non-output layers. Note that a random update would produce a 90 degree angle in expectation.

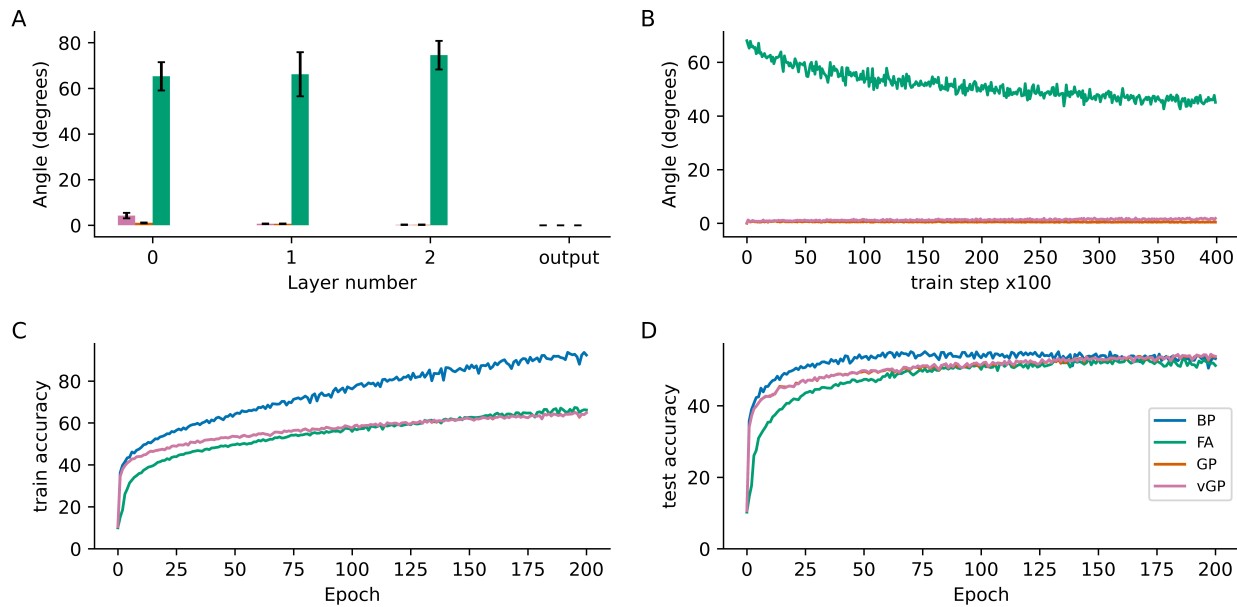

Figure 3: Performance of BP, vGP, GP and FA on CIFAR10 for a fully connected network with 3 hidden layers. For GP and FA, correspondence to BP's updates are displayed. A) Angles between GP's and FA's weight update vectors and BP's per layer. B) Angles in degrees between GP's and FA's updates and BP's weight updates per train step. C) Train accuracy BP, GP and FA per epoch. D) Test accuracy for BP, GP and FA per epoch.

Figure 3A shows that the angle is zero for the output layer for all algorithms. This is expected, as the updates are by definition identical for that layer. Updates in earlier layers in the network, which are arrived at after more iterations of (v)GP or FA, diverge very slightly in the case of GP and significantly in the case of FA. They are also almost identical in their test performance, as can be seen in Figure 3D, though BP stays ahead in train performance, as shown in Figure 3C. Note that the lines for GP (orange) and vGP (pink) overlap almost perfectly, though both are present in the figure. Though there is no gap in test performance, BP does seem to converge much more quickly on the train set. This likely reflects two things. (v)GP works on the assumption of perfectly orthogonal weight matrices, which are not present in practice, despite the regularizer being used. This means GP can only approximate BP's performance. Second, the presence of the orthogonality regularizer itself, dilutes (v)GP's task-related updates with orthogonality-related updates, slowing down convergence.

## 3.2 CIFAR10 results for a deep network

Figure 4 shows results for the same experimental setup, except with a deeper network, consisting of six hidden layers. As can be seen in Figure 4A and B, updates for vGP quickly diverge from BP's updates, with updates for GP maintaining their correspondence to BP for all layers. In terms of performance, vGP shows poor performance in this setting, while GP remains competitive with BP on the test set. FA outperforms vGP in this deeper network, but lags behind both BP and GP.

## 3.3 Deep convolutional networks

On the ImageNet dataset, like on CIFAR10, GP performs competitively with BP in terms of test performance, see Figure 5. FA almost completely fails to learn in this experiment, showing some learning only in the earliest epochs of training, followed by a decrease in performance as training continues. vGP was not included in the ImageNet experiment, as it failed to learn well even on the much simpler CIFAR10 task in a deep network.

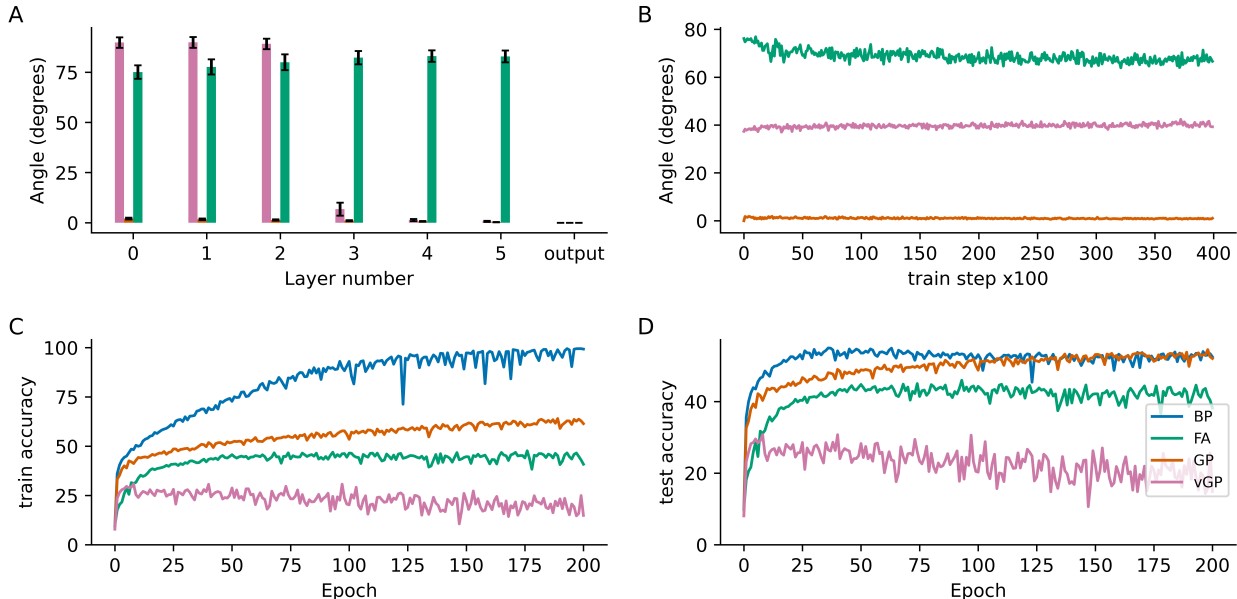

Figure 4: Performance of BP, vGP, GP and FA on CIFAR10 for a fully connected network with 6 hidden layers. For GP and FA, correspondence to BP's updates are displayed. A) Angles between GP's and FA's weight update vectors and BP's per layer. B) Angles in degrees between GP's and FA's updates and BP's weight updates per train step. C) Train accuracy BP, GP and FA per epoch. D) Test accuracy for BP, GP and FA per epoch.

Angles between GP's and BP's updates are again consistently low, though higher than in the CIFAR10 experiments. Angles between FA's and BP's updates fluctuate around 90 degrees, indicating little to no correspondence. Even for GP it can be seen that the updates slightly diverge from BP's in deeper layers, with the average angle at around 10 degrees, but reaching slightly over 20 degrees for the first hidden layer. It can also be seen in Figure 5 that GP converges slower than BP on the train set, though it eventually reaches parity in test accuracy. As in the CIFAR10 experiments, we attribute the gap in train performance to weight matrices not being perfectly orthogonal in practice and the orthogonality regularizer slowing down convergence. There is thus a tradeoff when using GP, where regularizing more strongly gives a better correspondence to BP at the cost of slower convergence.

The above interpretation was confirmed by running the network without performing any task-related weight updates, in which case the angles stay near zero throughout the network.

## 4    Discussion

This paper set out to investigate how to scale GP to more challenging problems. We found that in principle, GP scales to larger and more complex architectures than the relatively shallow fully connected networks used in previous research. This was achieved by tackling its numerical instability issues and adapting the algorithm to CNN architectures. Other biologically inspired learning rules either fail to scale to problems the size of ImageNet (Bartunov et al., 2018), require complex network topologies (Whittington & Bogacz, 2017; Millidge et al., 2022; Han et al., 2018; Payeur et al., 2021; Greedy et al., 2022) or require a slow iterative process to learn Scellier & Bengio (2017); Laborieux et al. (2021).

The main limitation of GP in terms of its performance is its reliance on invertible network architectures. This makes it incompatible with certain activation functions, as well as non-invertible operations like max-pooling.

Additionally, GP, in its current implementation, runs significantly slower than BP. This is due to the need to invert weight matrices, include auxiliary units, apply the orthogonality regularization and perform a

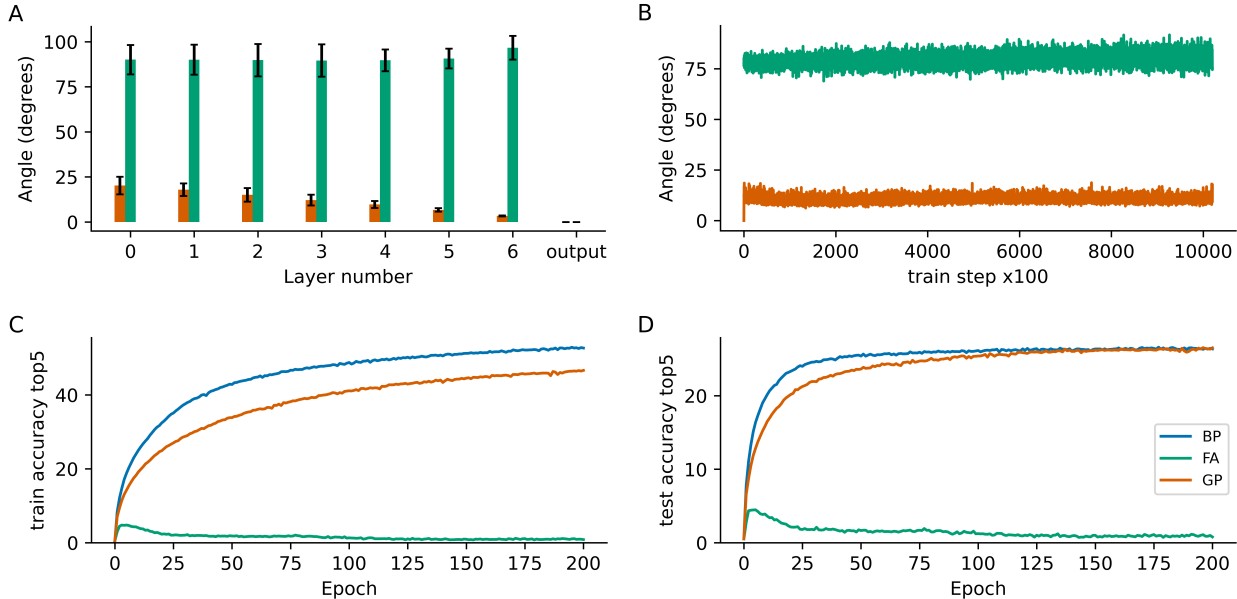

Figure 5: Performance and correspondence with BP for GP and FA on ImageNet in a convolutional neural network with 7 convolutional layers. A) Angles in degrees between BP's updates and those by GP and FA per layer. B) Angles in degrees between BP's updates and those by GP and FA per train step. C) Top 5 train accuracy for BP, GP and FA per epoch. D) Top 5 test accuracy for BP, GP and FA per epoch.

customized, and thus less optimized, backward pass. This slowdown relative to BP stems from the specifics of modern computer architectures. A biologically inspired algorithm such as GP would be expected to run much faster on neuromorphic hardware optimized for such computations compared to current hardware. Due to its lack of biological realism, BP likely would not run on such hardware at all, if this hardware is subject to the same physical constraints as the human brain, e.g. the locality of information.

Finally, the presence of an orthogonality regularizer and the fact that GP's assumptions are not fully realized in practice appear to slow down convergence relative to BP.

As discussed above, some of the remaining limitations may be remedied in various ways. The goal of the current research, however, is to demonstrate that, in principle, biologically plausible learning can scale to real-world problems. This is demonstrated by GP's competitive performance with BP on the complex ImageNet problem, given some constraints on the architecture. We argue that demonstrating the efficacy of any new learning algorithm to large-scale problems is crucial since algorithms that seemingly work well on toy problems have a tendency to break down in more complex settings.

A reader may also ask whether the issues addressed in this work are relevant for biology. For example, the issue overcome by normalization of the target outputs is an issue of precision of floating point numbers. However, we would argue that in biology, any noise floor would also ensure that a sufficiently small target signal would be drowned out amidst network activity (Faisal et al., 2008). One might also ask whether an orthogonality regularizer is biologically plausible, given that in order to orthogonalize weights, some non-local information needs to be used. Here, we would argue that lateral inhibition is a well-established phenomenon in neuroscience that could produce a similar decorrelating effect (Békésy, 1967). Therefore we propose that these improvements are also relevant to any biologically motivated analysis of these learning rules.

The presence of (approximately) inverse weights between layers and the need for separate forward and backward phases in the network seem to be the most prominent remaining departures from biological plausibility, making the elimination of these mechanisms excellent targets for future research.

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

## A  Performance comparisons

Table 1: Final train and test accuracies for networks trained on CIFAR10 (top-one accuracy) and ImageNet (top-five accuracy) using backpropagation (BP), normalized and vanilla GP (GP) and feedback alignment (FA).

| | CIFAR10 (shallow) | | CIFAR10 (deep) | | ImageNet (deep) | |
|---|---|---|---|---|---|---|
| **Algorithm** | **Train** | **Test** | **Train** | **Test** | **Train** | **Test** |
| BP | 92.4 | 53.2 | 99.2 | 52.5 | 52.7 | 26.4 |
| GP (normalized) | 64.9 | 53.7 | 61.4 | 52.0 | 46.6 | 26.5 |
| GP (vanilla) | 64.8 | 53.8 | 14.9 | 14.8 | - | - |
| FA | 66.2 | 51.3 | 40.8 | 38.2 | 0.8 | 0.8 |

## B  Pseudocode for normalized GP's backward pass

---
**Algorithm 1** Pseudocode for normalized GP's backward pass.
---
**parameters:** $t_L$, the network's activation target at the output layer; $a_l$, the network's activation from the forward pass for each layer; $G_l$, an inverse mapping for each network layer; $D_l$, the data dependent derivatives for each layer; $\eta$, a small constant; $\alpha$, a learning rate

**for** layer $l$ **from** $L$ **to** $0$ **do**

$\quad W_l \leftarrow W_l - \alpha D_l (a_l - t_l) a_{l-1}^\top$  $\qquad\qquad$ ▷ Compute weight update based on quadratic loss

$\quad b_l \leftarrow b_l - \alpha D_l (a_l - t_l)$  $\qquad\qquad\qquad\qquad$ ▷ Compute bias update

$\quad \gamma_l = \frac{\eta}{\|a_l - t_l\|^2}$  $\qquad\qquad$ ▷ Set perturbation factor to normalize target-activation differences

$\quad \epsilon_l = \gamma_l D_l^2$  $\qquad\qquad\qquad\qquad$ ▷ Adjust perturbation factor for local gradients

$\quad t_{l-1} = G_l \left( (I - \epsilon_l)\ a_l + \epsilon_l\ t_l \right)$  $\qquad\qquad$ ▷ Propagate targets back through network

**end for**

**return** updated weights $W_l$, updated biases $b_l$
---

## C  Pseudocode for inversion of a convolutional layer

---
**Algorithm 2** Inversion of convolutional layer
---
**parameters:** $y_l$, the convolutional layer's output; $w_l$, the convolutional layer's kernel; $x$, the input of the convolutional layer's forward pass; $n_x$ mapping that contains for each element of $x$ how many overlapping filters it is in the receptive field of

$\hat{x} = \texttt{Transposed\_Convolution}(y, w_l^\top)$

$\hat{x} \leftarrow \hat{x} - x \odot (n_x - 1)$

**return** reconstructed input $\hat{x}$
---

## D  Code availability

Code used for the experiments can be found here: https://github.com/artcogsys/GAIT_prop_scaling/

