# OpenReview forum: "Gradient-adjusted Incremental Target Propagation Provides Effective Credit Assignment in Deep Neural Networks"
_TMLR — Accepted by TMLR_

### Review · Reviewer_wyuX · 2022-10-25

**Summary Of Contributions:**

This paper extends and improves on a previously proposed biologically motivated alternative to backprop, called GAIT-prop, which is a variant on target propagation (that worked better than previous variants). The authors propose to normalize the coefficient used to mix the fed back targets and the feedforward activations when computing the new targets, inversely proportional to the distance between feedforward activations and the fed back targets.  They also propose a way to deal with weight matrices that are not full, such as the implicit matrices corresponding to convolutions. On the chosen architectures, results are better than the compared methods (feedback alignment and the previous GAIT-prop version, unnormalized) and results comparable to backprop with a particular architecture (and no deformations?) are obtained by the authors on ImageNet on the test set but still substantially worse on the training set and MUCH worse than even AlexNet or the current state-of-the-art in test performance.

**Audience:**

Yes

**Claims And Evidence:**

No

**Requested Changes:**

* as it stands, the title is overclaiming, since the actual results are MUCH worse than even the 2012 ImageNet state-of-the-art: either change the title (and the corresponding claims in the abstract, intro, discussions and conclusion) or obtain more credible results
* provide actual numbers of final accuracy in a table
* add parameter count comparisons and discuss the very weak results obtained on ImageNet, or better train larger models that can at least compete with backprop circa 2012
* add and discuss the missing references
* explain Eq. 4
* justify Eq. 5 better and discuss why there are no issues with epsilon_l > 1 (or if there are, discuss them)
* provide a clear and formal description of the method to deal with convolutions and demonstrate why it is the (or a) correct way to handle covolutional operators
* add pseudo-code for normalized GAIT-prop with convolutions
* clarify where requested (see above)
* use honest statements when discussing the figures, not blaming the inability of GAIT-prop to match training performance of backprop on the overfitting of backprop, and not claiming that GAIT-prop would eventually catch up when the evidence shown in the figures 4 and 5 suggests exactly the opposite.


**Strengths And Weaknesses:**

er is the shared weights (transposed) between the forward and the backward path (a.k.a the weight transport problem) and between the derivatives of the forward and backward "activation", however these joint computations would become "local" in the proposals where pyramidal neurons host both the forward and the backward computation.
* insert "test" in the first line of page 7 (before "performance"), to be honest, and change the statement "In terms of train accuracy, BP does seem to converge more quickly than the other algorithms" to something like "In terms of train accuracy, BP achieved much better performance much faster and the gap continued to grow by the end of training". Similarly, remove the disingenous statement in the last sentence of sec 3.
* the non-plausibility for biological networks of inverting matrices should not be only discussed in last paragraph of the paper but should be stated upfront in the introduction and section 2.1, e.g. around Eq. 3.
* The cartoon explanation in Fig 2 is very confusing because it looks like applying an inverse matrix to a small vector (the output vector) could recover a rich high-dimensional image (the image patch, 2nd row of the figure). This is of course not plausible unless the output dimension is huge (corresponding to H x W x C_in). Hence the figure looks wrong. It is only when reading the text that we understand that indeed C_out is chosen in this way. This should be clarified in the caption or the whole figure should be redesigned. In any case it did not really make it clear to me what the procedure was and why it worked.

Missing references:
There is a nice review of the litterature but it is missing what I consider some important papers, especially in the context of GAIT-prop:
  * Equilibrium Propagation (Scellier et al, Frontiers Comp Neuro 2017)
  * Sacramento et al NeurIPS 2018
  * several follow-up on EqProp including Laborieux et al (also Front. Comp Neuro, but 2021)
  and Scellier et al 2019 (Neural Comp)
  * Meulemans et al NeurIPS 2020, ICML 2022 and NeurIPS 2022 (accepted, but it is on arXiv)

---

### Review · Reviewer_iHJb · 2022-10-31

**Summary Of Contributions:**

The submission considers modification of GAIT-prop, a biologically plausible alternative to backprop based on Target Propagation, that help GAIT-prop scale to larger networks and datasets. The main contributions are improvements to numerical stability and the introduction of an inverse convolution.

**Audience:**

Yes

**Broader Impact Concerns:**

I don't foresee any broader impact concerns.

**Claims And Evidence:**

Yes

**Requested Changes:**

Some kind of plot that shows the scaling of the different methods with the full depth of the network would be very useful. For example, how does the final test/train accuracy scale with the depth of the network. (i.e. train with a sequence of deeper networks not just 2, and summarize the results in one or two plots)

I believe this is a good work in addressing the scaling properties of bio-plausible learning algorithms but with the current plots, it is demonstrated that the submission's proposal performs better than original GAIT-prop but the *scaling* itself has not become clear.

For other changes see weaknesses.

**Strengths And Weaknesses:**

Strenghts

- The scaling of bio-plausible algorithms to big networks/data is an important problem that is often ignored.
- The manuscript is readable and highlights the problems that are to be solved without getting too bogged down in details (but see weaknesses below).

Weaknessses

- A lot of the details are missing.
   -  To be self-contained, section 2 should be expanded to summarize the following (for detail the reader can be referred to the respective papers but a one or two sentence summary is good):
      - What is the local gradients problem?
      - how is the orthogonality constraint implemented in a bio plausible way.
      - how is $D(a_l)$ computed in a bio-plausible way? Doesn't $d F_l/ d a_l$ have a weight transport problem? It needs $W^\top.$
- In section 2.1 it is not fully clear in the text why local connectivity is a problem (it is only later that it is mentioned that the inversion is the issue).
- The exact methodology is hard to understand. It would be useful to have algorithm boxes such that the *exact* steps for both normalization and inversion can be understood.
- The plots can use error bars. Running the each algorithm a few times so that the reader understands the run to run variability of the different methods is very important.
- How does predictive coding scale with depth? The authors dismiss PC on realism grounds (error neurons) but this argument of what is more/less realistic is rather subjective. (I can see readers also doubting many of the choices made in GAIT-prop as plausible but unrealistic.) In this light, it would be useful for the authors to at least comment on the scaling properties of PC. (and ideally provide comparable experiments.) This would make the paper a lot more appealing as a work on how bio-plausible algorithms scale with depth. (Even if modified GAIT-prop ends up losing to PC in scaling, this would still be a very valuable contribution.)

---

### Review · Reviewer_r9jh · 2022-11-11

**Summary Of Contributions:**

This paper aims to extend the previously introduced GAIT-prop learning algorithm to deeper networks and to convolutional layers. They argue that GAIT-prop is a more biologically plausible learning mechanism that has been theoretically shown to approximate backprop but that it is necessary to show that this approach scales. Specifically, they introduce a modification to the GAIT-prop targets to normalise the magnitude of their difference to the true activity. By normalising, this alleviates an issue they identified with standard GAIT-prop where small target differences are prevented from propagating due to numerical precision and large differences from deviating the updates far from backprop. To extend the approach to convolutional layers, they note that the convolution operation can be viewed as a simple matrix multiplication that can be inverted provided the same conditions of square and orthogonal weight/kernel matrices are enforced. They introduce a way to invert these matrices in a patch-by-patch manner to greatly improve computational efficiency. Finally, they show that their new model’s updates align to backprop and is competitive with backprop’s test accuracies on the CIFAR10 and ImageNet datasets.



**Audience:**

Yes

**Broader Impact Concerns:**

No concerns.

**Claims And Evidence:**

Yes

**Requested Changes:**

Note: If some of the points below are not feasible, we will take this into account.

Major points:

•	In the introduction it seems the authors are incorrect in their summarisation of the work of Guerguiev et al. (2017). They state that the separation of neurons into multiple compartments resolves the implausibility of requiring symmetric feedback weights which is not the case. This is evidenced by the fact that this work uses fixed random feedback weights (i.e. feedback alignment) for many of their experiments to avoid the implausibilities of this issue. Additionally, they claim that no separate feedback pathway is used despite the fact that separate feedback connections are used. Finally, they state that the weight transport problem is a separate problem that is also addressed however in this case it is the same problem as the symmetric feedback weights and, again, this work does not provide a new solution to this problem.

•	It is stated that “Target Propagation addresses the weight transport problem, because it uses only local activity targets to achieve multi-layer learning.” I don’t believe this is true since TP requires knowledge (i.e. weight transport) of the feedforward weights to construct the feedback weight matrix as its inverse in order to generate the targets. It can be potentially argued that TP allows for an easier mechanism of learning the feedback weights to avoid weight transport, but this is a separate argument.

•	In the introduction it says “no biologically plausible learning algorithm has so far managed to scale to ImageNet and remain competitive with BP”. While it is not competitive with BP it should be mentioned that Burstprop (Payeur et al. Nature Neurosci 2021) has been shown to scale to ImageNet. Moreover, a new model (BurstCCN, Greedy et al. NeurIPS 2022) is shown to align near-perfectly with backprop under certain conditions (see their CIFAR-10 task results). As BurstCCN builds on Burstprop it would imply that it would do well in Imagenet. Importantly, BurstCCN is currently the model that gets the closest to biology, capturing a wide range of experimental observations. These models should be discussed, as they are currently the best biologically plausible models of backprop-like credit assignment.

•	In Section 2.2.2 it is not immediately clear or explained how the reshaping of the 4d kernel matrix to be square is being accomplished. It would be useful to explain more clearly what the shape of the reshaped kernel is after reshaping but before transposing and inversing. Also, in Figure 2, is “image patch” referring to for example a 3x3 patch (with kernel size 3x3)? If so I think it would be useful to replace the full image that is being shown there with just the small patch to avoid confusion.

•	The plots in Figures 4-6 appear to be running with just a single seed even for the simpler CIFAR-10 task. It would be useful here to include the mean and variance of these quantities across multiple seeds.

•	Due to the large disparity between the training and test accuracy in all tasks for BP it would be important to discuss the use of regularisation in more detail. Firstly, does the orthogonality regulariser for used in GAIT-prop also impact generalisation in a similar way to an L1 or L2 regulariser? Secondly, why were other regularisation techniques not included (especially in BP) if it appears to be overfitting the training data significantly? Can they be successfully applied to GAIT-prop in combination with the orthogonality regulariser?

•	It is stated multiple times that normalised GAIT-prop is competitive with BP (in terms of test performance) but the differences in training performance seem to indicate that BP is still significantly more powerful but simply failing to generalise, would be important to make this clear.

•	Why are the models not trained until convergence in Figure 6?

Minor points:
•	In the introduction it is stated that “there is no explicit theoretical relationship between TP and BP” but also “there exists a direct correspondence between target-based learning and BP” which appear to contradict each other without a distinction being made more clear.

•	For Figures 4-6 it would be more clear if these were imported in a vector format (svg/pdf) instead of a png. Alternatively, a higher resolution png could be used.

•	For clarity, Fig. 6 should use the same colour for the “GAIT-prop normalised” as used in Fig. 5 and 4.

•	In Section 3.3, Layer 0’s update angle is referred to as the angle for the input layer but this is actually the first hidden layer’s update angle.

•	In Section 4, it is mentioned that normalised GAIT-prop does not use non-local information, but the layer-wide normalisation is arguable non-local information to the neuron. This point could be clarified that it is local to the layer.

•	“A biologically inspired algorithm such as GAIT-prop would be expected to run much faster on neuromorphic hardware optimized for such computations.”. Should clarify if this is in faster in reference to BP or just the currently used implementation.

•	In Section 4, it is argued that BP would likely not run on neuromorphic hardware at all. It would be useful to provide a reference for this claim.

•	In Section 4, it would be useful to include more insight into the link between the decorrelating effect of lateral inhibition and the orthogonality regularisation. The plausibility of this mechanism seems vital to the plausibility of the whole system.

•	Typos:
o	In Section 2.3 “hyper parameters”
o	Section 4: “GAIT-prop scales to larger and more complex architectures than the relatively shallow fully connected networks used in previous research”
o	Section 4: “algorithm that seemingly work”


**Strengths And Weaknesses:**

Overall, this is an important contribution to the field, which tries to scale up existing biologically plausible algorithms. But there are several points that should be ideally addressed/clarified as detailed below.

---

### Decision · Action_Editors · 2022-12-16

**Recommendation:** Accept as is

**Comment:**

The reviewers brought up several points to be addressed in the first review round. All reviewers were fully satisfied with the revision. The reviewers noted that the manuscript provides an important contribution to the field. On the negative side, it was noted that the performance is still much worse than standard optimization with backprop. However, this is a general characteristic of any more biologically plausible algorithm available to date.
  Another point was that the biological plausibility of exact inverses is doubtful, which reduces the significance from a biological point of view. Nevertheless, the reported progress it is still worth publishing.

**Audience:**

There is quite some interest in biologically plausible learning paradigms for neural networks. Although this is not the mainstream in machine learning, I believe a substantial part of the TMLR audience will be interested in the findings.

**Claims And Evidence:**

The manuscript extends GAIT-prop, a biologically motivated alternative to backprop. They claim that his improves performance of the algorithm. This is well supported by experiments.